# Clinical performance evaluation of TAQPATH Enteric Bacterial Select Panel for the detection of common enteric bacterial pathogens in comparison to routine stool culture and other qPCR-based diagnostic tests

Jasmin Koeffer,[1] Melissa Kolb,[1] Oceane Sorel,[2] Camilla Ulekleiv,[2] Jelena D. M. Feenstra,[2] Ulrich Eigner[1]

**ABSTRACT**  The use of real-time PCR-based methods for the detection of enteric pathogens in routine clinical diagnostics is slowly replacing stool culture, with open questions regarding clinical performance and ease of implementation. We evaluated the clinical and analytical performance of the TAQPATH Enteric Bacterial Select Panel in comparison to routine stool culture and other multiplex diagnostic tests. Clinical stool specimens ($N$ = 217) from symptomatic individuals were tested using the TAQPATH Enteric Bacterial Select Panel and the BD MAX Enteric Bacterial Panel, while the BIOFIRE FILMARRAY Gastrointestinal (GI) Panel was used as a resolver. This analysis demonstrated that the TAQPATH Enteric Bacterial Select Panel had clinical sensitivity and specificity of 100.00% for both *Salmonella* spp. and *Shigella* spp./Enteroinvasive *Escherichia coli* (EIEC) and 100.00% and 98.09% for *Campylobacter*, respectively. Analytical comparison of six European conformity-in vitro diagnostic (CE-IVD) tests on contrived samples showed similar performance results. In a prospective study, we tested 500 stool samples using the TAQPATH Enteric Bacterial Select Panel and routine stool culture combined with *Campylobacter* antigen test. In total 26/500 samples were positive for *Campylobacter*, 2/500 for *Salmonella,* and none for *Shigella*/EIEC using the TAQPATH kit. Routine culture and antigen testing detected only 65.4% (17/26) of *Campylobacter* infections. The results obtained by the TAQPATH kit were confirmed as true positive for *Campylobacter* using two resolver methods, indicating a significantly higher clinical sensitivity over routine stool culture. Finally, we demonstrated that the TAQPATH Enteric Bacterial Select Panel provided shorter time-to-result (<3 h vs 2–4 days), required sevenfold less hands-on time and 7.17-fold less laboratory plastic in comparison to conventional stool culture in routine diagnostics.

**IMPORTANCE**  Enteric bacterial infections caused by *Salmonella, Shigella,* pathogenic *Escherichia coli*, *and Campylobacter* represent one of the most common causes of infectious enteritis worldwide. The timely and accurate diagnosis of pathogens causing gastroenteritis is crucial for patient care, public health, and disease surveillance. While stool culture has long been the standard and highly specific method for detecting enteric pathogens, it is labor-intensive and time-consuming with limited sensitivity. To improve patient outcomes, there is a need to implement new cost-effective approaches for the detection of bacterial enteric pathogens with higher sensitivity and faster time to result. This study shows that multiplex real-time polymerase chain reaction-based tests, such as the TAQPATH Enteric Bacterial Select Panel, are accurate and cost-effective diagnostic alternatives for the detection and differentiation of the most common enteric bacterial pathogens, offering quicker time to result and higher sensitivity compared to routine stool culture.

Address correspondence to Ulrich Eigner, ulrich.eigner@labor-limbach.de.

O.S., J.D.M.F., and C.U. are employees of Thermo Fisher Scientific.

See the funding table on p. 12.

**KEYWORDS** molecular diagnostics, gastrointestinal infections, *Campylobacter*, *Salmonella*, *Shigella*

Bacterial enteric pathogens cause a significant burden of disease and death worldwide each year. The most common bacteria responsible for infectious gastroenteritis include *Salmonella, Shigella, pathogenic Escherichia coli* (*E. coli*), *and Campylobacter* (1–3). The identification of pathogens causing gastroenteritis is crucial for patient care, public health, and disease surveillance. Historically, the standard method for detecting bacterial pathogens in stool samples of infected individuals is routine stool culture. While culture is considered to be a highly specific method for detecting enteric pathogens, it is labor-intensive and time-consuming. It requires an incubation period of up to 3 days to allow the bacteria to grow to a detectable level before being able to identify the resulting colonies (4). This prolonged turnaround time can lead to significant diagnostic delays and initiation of appropriate treatment (5–14). In addition, some bacterial pathogens, including *Campylobacter* spp., have very specific and challenging culture requirements and can be difficult to grow, leading to low sensitivity, and therefore cases can be underdiagnosed (15). There is a need to implement new cost-effective approaches for the detection of bacterial enteric pathogens with higher sensitivity and faster time to result.

As an alternative, nucleic acid amplification tests such as multiplex real-time polymerase chain reaction (PCR)-based tests have recently been developed. These methods allow rapid, highly specific, and simultaneous detection of multiple pathogens in a single PCR reaction reducing the time to detection to as little as a few hours while optimizing the use of laboratory resources (15, 16). In addition, recent studies have shown that real-time PCR can detect pathogens with higher sensitivity, allowing for the detection of infections that may be missed by culture (17–24).

TAQPATH Enteric Bacterial Select Panel is a multiplex real-time PCR test designed for the detection of *Salmonella* spp*., Shigella* spp*./*Enteroinvasive *Escherichia coli* (EIEC), and several *Campylobacter* species (*Campylobacter coli, Campylobacter upsaliensis,* and *Campylobacter jejuni*). The goal of this study was to evaluate the clinical performance of the TAQPATH Enteric Bacterial Select Panel in comparison to another multiplex real-time PCR test, the BD MAX Enteric Bacterial Panel, as well as routine stool culture. In addition, we performed an analytical performance comparison of six different CE-IVD tests for the detection of *Shigella*, *Salmonella*, and *Campylobacter* in contrived stool samples. Finally, we assessed the effectiveness of implementation, time-to-result, and consumption of laboratory plastics between the routine stool culture and molecular diagnostic testing using the TAQPATH Enteric Bacterial Select Panel.

## MATERIALS AND METHODS

### Clinical samples

Stool specimens representing leftovers from routine diagnostic testing from a total of 717 individuals who exhibited symptoms of acute gastroenteritis were included in this study. In the first arm of the study, a total of 217 stool samples were collected from March 2019 to October 2022 in Germany, France, and Ivory Coast. These specimens were stored at −80°C before being included in the performance evaluation of the TAQPATH Enteric Bacterial Select Panel against the BD MAX Enteric Bacterial Panel (Fig. 1). Samples were thawed at room temperature and diluted in PBS at the volume ratio of 1:4 before being analyzed using molecular tests. For long-term storage, leftover samples were stored in PBS at −80°C. Clinical information was available for 167/217 samples. The median patient age was 40 years old (range: 2 months to 90 years). Overall, 60.5% of patients were female and 39.5% were male. The routine culture results indicated that the samples included the following strains of bacteria: *Campylobacter coli, Campylobacter jejuni, Salmonella enterica* Serotype Aarhus*, Salmonella enterica* Serotype Enteritidis*, Salmonella enterica* Serotype Indiana*, Salmonella enterica* Serotype Javiana*, Salmonella enterica*

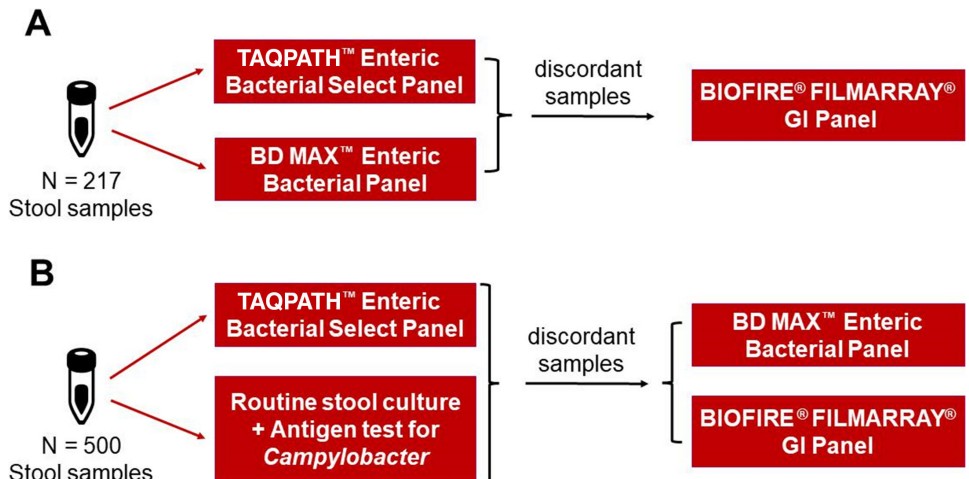

**FIG 1** Study design for the performance comparison of TAQPATH Enteric Bacterial Select Panel against (A) BD MAX Enteric Bacterial Panel; and (B) routine stool culture combined with an antigen test for *Campylobacter* detection.

Serotype Newport*, Salmonella enterica* Serotype Ohio*, Salmonella enterica* Serotype Typhimurium*, Salmonella enterica* Serotype 4,5,12:i:–*, Shigella flexneri, Shigella sonnei*, and *Salmonella* spp. The second arm of the study included a total of 500 stool samples collected in the period of January–March 2023 at Laboratory Limbach Heidelberg, Germany. They were used to assess the performance of the TAQPATH Enteric Bacterial Select Panel in comparison to routine stool culture combined with antigen testing for *Campylobacter* spp. (Fig. 1). All specimens included in the study were deidentified and assigned a study identification number. In this arm of the study, 50% male and 50% female patients were included. The median patient age was 68 years (range: 3 weeks to 97 years old). The bacteria strains identified in these patients by routine stool culture were *Campylobacter coli, Campylobacter jejuni, Salmonella* Typhimurium, and *Salmonella* Weltevreden.

## Real-time PCR testing of clinical stool samples

All stool specimens included in the study were tested with the TAQPATH Enteric Bacterial Select Panel (Thermo Fisher Scientific) according to the instructions for use. The nucleic acid extraction was performed using the MagMAX Microbiome Ultra Nucleic Acid Isolation Kit on the Microbiome Bead Plate and the King Fisher Flex and bead beater instruments, while realtime PCR was performed using the QuantStudio 5 Real-Time PCR system. The retrospective study performed on 217 samples included parallel testing using the BD MAX Enteric Bacterial Panel (Becton Dickinson) assay, according to the manufacturer's instructions for use on the BD MAX system. Samples showing discordant results between the TAQPATH Enteric Bacterial Select Panel and the BD MAX Enteric Bacterial Panel or routine stool culture were tested using the BIOFIRE FILMARRAY Gastrointestinal (GI) Panel (bioMérieux) assay on the BIOFIRE FILMARRAY torch system according to the instructions for use.

## Bacterial stool culture

To isolate *Salmonella* spp. and *Shigella* spp., all stool specimens were inoculated onto MacConkey agar (bioMérieux), Hektoen agar, and Selenit-F Broth (Becton Dickinson) media. Culture plates were then incubated at 35℃ ± 2℃ for 48 hours and examined after 24 and 48 hours. After 24 hours of incubation, subculture was performed from enrichment Selenit-F broth media to both Chromid Salmonella Elite media (bioMérieux) and Xylose-Lysine-Deoxycholate-agar media (Becton Dickinson) with a sterile loop. Culture plates were incubated for 24 hours at 35℃ ± 2℃. Colonies suspected

of being *Salmonella* spp. were identified using matrix-assisted laser desorption/ionization-time of flight mass spectrometry (MALDI-TOF MS) (MALDI Biotyper Microflex or Sirius system, Bruker Daltonics). *Salmonella* serovars were further determined using specific antisera (Sifin Diagnostics, Berlin; Bio-Rad Laboratories, Hercules, CA, USA). Suspected *Shigella* spp. colonies were inoculated onto Kligler iron agar media (Oxoid–ThermoFisher Scientific, Basingstoke, UK). After incubation for 18–24 hours at 35°C ± 2°C, the Kligler test media was read for characteristic *Shigella* reactions. Genus identification was performed through slide agglutination using commercially available *Shigella* antisera (Sifin Diagnostics) and biochemical procedures (api20 and apiCHE, bioMérieux). To isolate *Campylobacter* spp., all stool specimens were inoculated onto *Campylobacter* selective agar (Becton Dickinson) and incubated under microaerophilic conditions at 42°C for 3 days. Suspected *Campylobacter* spp. colonies were confirmed by oxidase test and MALDI-TOF MS (MALDI Biotyper Sirius, Bruker Daltonics).

## Comparative evaluation of limit of detection

Contrived samples were created by inoculating cultures of *C. jejuni*, *S.* Typhimurium, and *S. sonnei* individually into negative stool samples at two concentrations (~3–5× limit of detection, LOD and 10× LOD of the TAQPATH Enteric Bacterial Select Panel) in 15 replicates. The respective bacterial concentrations were adjusted after diluting a 0.5 McFarland solution. The turbidity according to McFarland was adjusted in saline using a densitometer (DensiCHEK Plus, bioMérieux). The LOD of the TAQPATH Enteric Bacterial Select Panel were the following: for *Salmonella* Typhimurium 500 CFU/mL , for *C. jejuni* 66 CFU/mL, and for *S. sonnei* 60 CFU/mL. Contrived positive samples and 30 negative samples were tested in parallel using six CE-IVD kits: TAQPATH Enteric Bacterial Select Panel, RIDAGENE Bacterial Stool Panel I, BD MAX Enteric Bacterial Panel, Allplex GI-EB (Seegene), LightMix modular CE-IVD assays (Roche/TiB MolBiol), and BIOFIRE FILMARRAY Gastrointestinal Panel. All tests were performed according to their respective instructions for use. Testing using the RIDAGENE Bacterial Stool Panel I was performed using the MagNa Pure 96 System for extraction and the LightCycler 480II for PCR. The Allplex GI-EB assay (Seegene) was performed using the Seegene STARlet System and the CFX96 real-time PCR detection system. LightMix modular CE-IVD assays (Roche/TiB MolBiol) were performed on the LightCycler 480 II instrument following the nucleic acid extraction using the Small Volume Kit on the MagNa Pure 96 System. Testing using the TAQPATH Enteric Bacterial Select Panel, BD MAX Enteric Bacterial Panel, and BIOFIRE FILMARRAY Gastrointestinal Panel was performed as described for the clinical stool specimens.

## Effectiveness of implementation analysis

To assess the effectiveness of implementing the TAQPATH Enteric Bacterial Select Panel compared to routine stool culture, the cost of consumables per type of test, time-to-result including result interpretation, hands-on time, and amount of all laboratory plastic consumables (including transfer pipettes, pipette tips, tubes, plastic foils, loops, PCR plates, Micro titer plate ELISA, and petri dishes measured in grams) used in the workflow was compared between the TAQPATH Enteric Bacterial Select Panel and routine stool culture. The time-to-result was measured as the mean time between the moment a sample was taken for processing at the laboratory to the final conclusive result of the analysis.

## Statistical analysis

Performance comparison included calculations of positive and negative percent agreements. Negative percent agreement (NPA) was calculated as the ratio between the number of negative results by both the evaluation method and its comparator and the number of negative results by the comparator method multiplied by 100. Positive percent agreement (PPA) was calculated as the ratio between the number of positive

results by both the evaluation method and its comparator and the number of positive results by the comparator method multiplied by 100. Following discordant sample resolution by the resolver method in which the result obtained by two out of three tests was considered as the true result, the clinical sensitivity and specificity were calculated. The clinical specificity was calculated as the ratio of the number of true negative results divided by the number of true negative plus false positive results multiplied by 100. The clinical sensitivity was calculated as the ratio of the number of true positive results divided by the number of true positive plus false-negative results multiplied by 100. The two-sided 95% confidence intervals were calculated using the Clopper-Pearson method.

## RESULTS

### Clinical performance evaluation of the TAQPATH Enteric Bacterial Select Panel in comparison to the BD MAX Enteric Bacterial Panel

First, a retrospective study was performed on a total of 217 stool samples collected from individuals with symptoms of acute gastroenteritis. All samples were tested in parallel using the TAQPATH Enteric Bacterial Select Panel and BD MAX Enteric Bacterial Panel (Fig. 1). Two samples were excluded from further analysis due to repeated invalid results in the BD MAX System. The analysis demonstrated high concordance between the two multiplex real-time PCR kits, with PPA and NPA of >96% for all targets (Table 1). The positive sample cohort spanned the dynamic range of the assays with low, medium, and high burden of pathogens as measured by the Ct values obtained (Fig. 2). Up to 45% of the positive samples had high Ct values (Ct ≥ 30), which showed that both assays exhibited excellent sensitivity including for samples with lower bacterial load (Fig. 2). A total of six discordant samples were identified (Table 2) and were further analyzed using the BIOFIRE FILMARRAY GI Panel. Of the five samples showing positive results for *Campylobacter* spp. using the TAQPATH Enteric Bacterial Select Panel, two samples were confirmed as positive with the BIOFIRE FILMARRAY GI Panel. Of note is that another 2/5 samples that showed a positive result using the TAQPATH Enteric Bacterial Select Panel test and negative result on the comparator and resolver PCR assays were initially included in the study as *Campylobacter* spp. positive samples as detected by routine stool culture. For additional samples showing discordant results for *Salmonella* spp. and *Shigella* spp./EIEC, all results obtained from the resolver were in accordance with the results of the TAQPATH Enteric Bacterial Select Panel. Upon discordant sample resolution using the resolver molecular method, the clinical sensitivity and specificity of the TAQPATH Enteric Bacterial Select Panel were 100.00% for both *Salmonella* spp. and *Shigella* spp./EIEC, while for *Campylobacter* spp., the sensitivity and specificity were 100.00% and 98.09%, respectively (Table 3).

### Analytical performance comparison of six CE-IVD tests for the detection of *S. sonnei*, *C. jejuni*, and *S.* Typhimurium

To compare the analytical performance of six CE-IVD tests for the detection of *Shigella sonnei*, *Campylobacter jejuni*, and *Salmonella* Typhimurium, samples were contrived by spiking live organisms into negative stool sample in two different concentrations representing the 3–5× and 10× the limit of detection of the TAQPATH Enteric Bacterial Select Panel. This analysis demonstrated comparable analytical sensitivity between the TAQPATH Enteric Bacterial Select Panel, the BIOFIRE FILMARRAY Gastrointestinal Panel, and the RIDAGENE Bacterial Stool Panel I, which was higher for *C. jejuni* and *S.* Typhimurium compared to the Allplex GI-EB screening kit, LightMix modular CE-IVD assays (Roche/TiB MolBiol), and the BD MAX Enteric Bacterial Panel (Fig. 3). The results also showed comparable performance for the detection of *S. sonnei* for all tests except for the BD MAX Enteric Bacterial Panel and the LightMix modular CE-IVD assays (Roche/TiB MolBiol), which demonstrated lower sensitivity (Fig. 3). The difference in the performance of the tests observed reflected the difference in the LOD of the different diagnostic tests.

**TABLE 1** Performance comparison of TAQPATH Enteric Bacterial Select Panel to BD MAX Enteric Bacterial Panel for the detection of *Campylobacter* spp., *Salmonella* spp., and *Shigella* spp./EIEC

| | | BD MAX Enteric Bacterial Panel | | | | | | | | |
| --- | --- | --- | --- | --- | --- | --- | --- | --- | --- | --- |
| | | *Campylobacter* spp. | | | *Salmonella* spp. | | | *Shigella* spp./ EIEC | | |
| | | Positive | Negative | Total | Positive | Negative | Total | Positive | Negative | Total |
| TAQPATH | Positive | 56 | 5 | 61 | 57 | 1 | 58 | 31 | 1 | 32 |
| Enteric | Negative | 0 | 154 | 154 | 1 | 156 | 157 | 0 | 183 | 183 |
| Bacterial Select Panel | Total | 56 | 159 | 215 | 58 | 157 | 215 | 31 | 184 | 215 |
| Positive percent agreement (95% CI) | | 100.00 (93.58%–100.00%) | | | 98.28 (90.86%–99.70%) | | | 100.00 (88.97%–100.00%) | | |
| Negative percent agreement (95% CI) | | 96.86 (92.85%–98.65%) | | | 99.36 (96.48%–99.89%) | | | 99.46 (96.99%–99.90%) | | |

## Clinical performance evaluation of the TAQPATH Enteric Bacterial Select Panel in comparison to the routine stool culture combined with an antigen test

To evaluate the clinical performance of the TAQPATH Enteric Bacterial Select Panel in comparison to the routine diagnostic procedure based on the stool culture and RIDASCREEN *Campylobacter* antigen test, a prospective study was performed on 500 additional stool samples collected from January to March 2023 in Germany from patients experiencing symptoms of acute gastroenteritis (Fig. 1). All 500 samples were tested in parallel using the PCR-based approach and the routine culture combined with an antigen test. PPA and NPA between the two methods were calculated (Table 4). No positive samples for *Shigella* spp./EIEC were detected in the sampled population; hence, no PPA could be calculated, while only two individuals tested positive for *Salmonella* spp. by both methods (Table 4). *Campylobacter* spp. was detected in 26/500 samples with the TAQPATH Enteric Bacterial Select Panel, while routine culture combined with antigen testing detected only 65.4% (17/26) of these infections, resulting in a PPA and NPA of 94.44% and 98.13%, respectively (Table 4). Within the 17 samples detected as positive with the combination of routine stool culture and an antigen test, five samples were detected as positive with the antigen test only, while the culture was negative. Discordant samples for *Campylobacter* spp. were tested using two resolver methods including the BD MAX Enteric Bacterial Panel and the BIOFIRE FILMARRAY GI Panel. The results obtained with the TAQPATH Enteric Bacterial Select Panel were shown to be true positive for *Campylobacter* spp. using either the BD MAX Enteric Bacterial Panel or the BIOFIRE FILMARRAY GI Panel, meaning that the method using a combination of routine stool culture and antigen testing failed to detect *Campylobacter* spp. in 34.6% (9/26) of samples (Table 5). Based on the Ct values obtained, the discordant samples did not show low bacterial DNA load. Given that the BD MAX Enteric Bacterial Panel is designed to only detect *Campylobacter jejuni* and *Campylobacter coli* while the TAQPATH Enteric

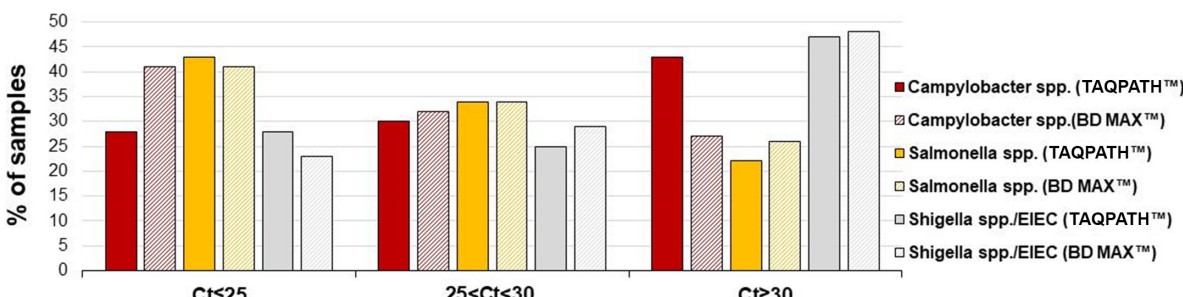

**FIG 2** Distribution of Ct values for the positive cohort for each organism based on TAQPATH Enteric Bacterial Select Panel and BD MAX Enteric Bacterial Panel results.

**TABLE 2** Discordant sample resolution using BIOFIRE FILMARRAY GI Panel as a resolver[a]

| Sample ID | Culture | TAQPATH Enteric Bacterial Select Panel | | | BD MAX Enteric Bacterial Panel | | | BIOFIRE FILMARRAY GI Panel | | |
|---|---|---|---|---|---|---|---|---|---|---|
| | Campylobacter/ Salmonella/Shigella | Campylobacter spp. | Salmonella spp. | Shigella spp./ EIEC | Campylobacter spp. | Salmonella spp. | Shigella spp./ EIEC | Campylobacter spp. | Salmonella spp. | Shigella spp./ EIEC |
| GI_MSA_01_099 | Campylobacter | Positive (35.27) | Negative | Negative | Negative | Negative | Negative | Negative | Negative | Negative |
| GI_MSA_01_167 | Campylobacter | Positive (35.8) | Negative | Negative | Negative | Negative | Negative | Negative | Negative | Negative |
| GI_MSA_01_200 | N/A | Positive (36.34) | Negative | Positive (23.97) | Negative | Positive (38.3) | Positive | Negative | Negative | Positive |
| GI_MSA_01_082 | N/A | Positive (29.34) | Positive (35.44) | Positive (29.18) | Negative | Negative | Positive | Positive | Positive | Positive |
| GI_MSA_01_117 | N/A | Negative | Negative | Positive (33.39) | Negative | Negative | Negative | Negative | Positive | Positive |
| GI_MSA_01_054 | Campylobacter | Positive (32.99) | Negative | Negative | Negative | Negative | Negative | Positive | Negative | Negative |

[a]The Ct value is indicated in brackets. The column "Culture" shows results obtained by routine stool culture prior to the inclusion of the samples in the study. N/A, not applicable.

**TABLE 3** Clinical sensitivity and specificity of the TAQPATH Enteric Bacterial Select Panel

| | TAQPATH Enteric Bacterial Select Panel | | |
| --- | --- | --- | --- |
| | *Campylobacter* spp. | *Salmonella* spp. | *Shigella* spp./EIEC |
| Clinical sensitivity (95% CI) | 100.00% (93.79%–100.00%) | 100.00% (93.79%–100.00%) | 100.00% (89.28%–100.00%) |
| Clinical specificity (95% CI) | 98.09% (94.53%–99.35%) | *100.00% (97.61%–100.00%)* | 100.00% (97.94%–100.00%) |

Bacterial Select Panel targets *Campylobacter upsaliensis* in addition to *C. jejuni* and *C. coli*, the four samples showing negative results with the BD MAX and culture but positive results with the BIOFIRE FILMARRAY GI Panel and the TAQPATH Enteric Bacterial Select Panel are likely to represent *Campylobacter upsaliensis* (Table 5). After the resolution of discrepant results, clinical sensitivity and specificity of the TAQPATH Enteric Bacterial Select Panel in this evaluation were calculated to be 100.00% and 99.16%, respectively, for *Campylobacter* spp.

## Effectiveness of implementation of the TAQPATH Enteric Bacterial Select Panel in comparison to routine stool culture

To assess the effectiveness of implementing the real-time PCR-based TAQPATH Enteric Bacterial Select Panel test into a laboratory in comparison to routine stool culture combined with antigen testing, several parameters were assessed. First, we measured the consumption of laboratory plastics, and thus plastic waste, generated during the entire workflow from sample to result for the TAQPATH Enteric Bacterial Select Panel test and routine stool culture. The total amount of plastics was determined to be 1,349.6 g for the molecular approach vs 9,672 g for the routine culture (excluding antigen testing) for 93 samples. This indicated that the use of the TAQPATH Enteric Bacterial Select Panel in the routine diagnosis of the most common bacterial enteric pathogens, required 7.17-fold less laboratory plastic, compared to stool culture. In addition, the hands-on time required to process the sample and the total time-to-result were compared for both approaches. For TAQPATH Enteric Bacterial Select Panel, processing of 93 samples required 59 minutes of hands-on time with a total time-to-result of 2 h and 48 minutes including sample dilution, DNA extraction, preparation of real-time PCR, real-time PCR, and data analysis. In comparison, the total hands-on time for routine stool culture for 93 samples, assuming that there were six positive samples, was 7 h and 13 minutes, with a total time to result of 2–4 days, indicating a significant advantage in results availability with the use of the molecular diagnostic test. In terms of consumable costs for the detection of *Salmonella* spp., *Shigella* spp./EIEC, and *Campylobacter*, the price per sample for the molecular approach including nucleic acid extraction and CE-IVD real-time PCR

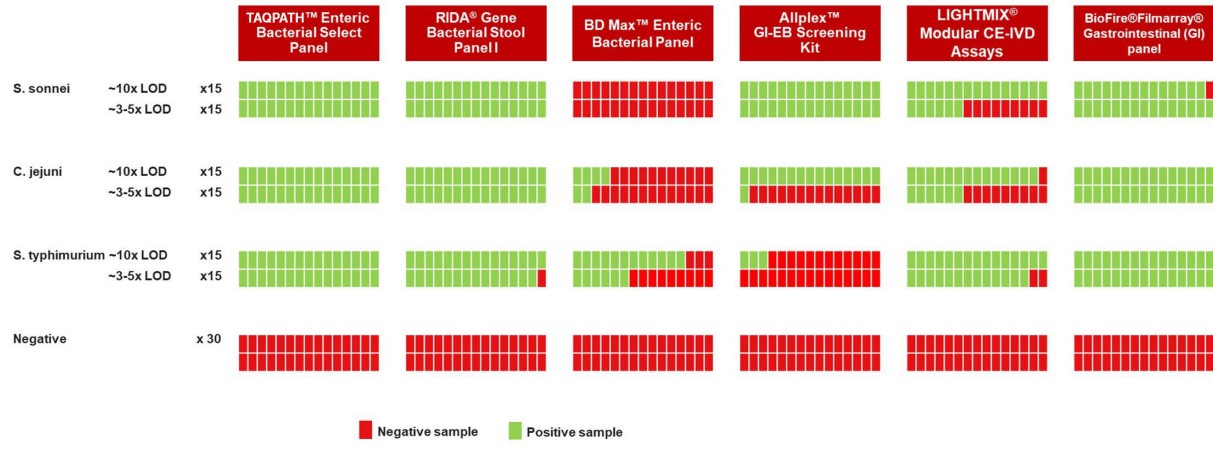

**FIG 3** Comparative evaluation of analytical performance for several CE-IVD tests on contrived samples for the detection of *S. sonnei*, *C. jejuni*, and *S.* Typhimurium.

**TABLE 4** Performance comparison between TAQPATH Enteric Bacterial Select Panel and routine stool culture combined with RIDASCREEN *Campylobacter* antigen test

| | | Routine stool culture + RIDASCREEN *Campylobacter* antigen test | | | | | | | | |
| | | *Campylobacter* spp. | | | *Salmonella* spp. | | | *Shigella* spp./ EIEC | | |
| | | Positive | Negative | Total | Positive | Negative | Total | Positive | Negative | Total |
|---|---|---|---|---|---|---|---|---|---|---|
| TAQPATH | Positive | 17 | 9 | 26 | 2 | 0 | 2 | 0 | 0 | 0 |
| Enteric | Negative | 1 | 473 | 474 | 0 | 498 | 498 | 0 | 500 | 500 |
| Bacterial | Total | 18 | 482 | 500 | 2 | 498 | 500 | 0 | 500 | 500 |
| Select | | | | | | | | | | |
| Panel | | | | | | | | | | |
| Positive percent agreement (95% CI) | | 94.44 (74.24%–99.01%) | | | 100.00 (34.24%–100.00%) | | | N/A[a] | | |
| Negative percent agreement (95% CI) | | 98.13 (96.49%–99.01%) | | | 100.00 (99.23%–100.00%) | | | 100.00 (99.24%–100.00%) | | |

[a]not applicable.

test was equal to the cost of routine stool culture per positive sample at about 25 euros, while the routine stool culture consumables were 30% cheaper per a negative sample. This analysis, however, did not include the cost of the specialized culture of *Campylobacter upsaliensis*, which would be detected in the molecular approach, the phenotypical differentiation of serotypes, and the technical time used by technicians for sample processing, which is sevenfold higher for routine stool culture, in addition to the lower sensitivity the culture approach offered for the detection of *Campylobacter* infections.

## DISCUSSION

In this study, we showed that the TAQPATH Enteric Bacterial Select Panel can detect *Shigella*, *Salmonella*, and *Campylobacter* with high sensitivity and specificity in stool samples when compared to other available molecular diagnostic real-time PCR-based multiplex assays as well as routine stool culture. In addition, we demonstrated that the use of TAQPATH Enteric Bacterial Select Panel can reduce the consumption of laboratory plastics, hands-on time, and time-to-result compared to routine culture,

**TABLE 5** Resolution of samples with discordant results for *Campylobacter* spp. obtained with routine stool culture combined with RIDASCREEN antigen test and the TAQPATH Enteric Bacterial Select Panel

| Sample ID | TAQPATH Enteric Bacterial Select Panel | | Routine stool culture | RIDASCREEN antigen test | BD MAX Enteric Bacterial Panel | BIOFIRE FILMARRAY GI Panel |
| | Result | Ct value | | | | |
|---|---|---|---|---|---|---|
| GI_MSA_02_035 | Negative | —[b] | Positive | Negative | Negative | N/A |
| GI_MSA_02_030 | Positive | 19.36 | Negative | Negative | Negative | Positive |
| GI_MSA_02_486 | Positive | 26.64 | Negative | Negative | Negative | Positive |
| GI_MSA_02_064 | Positive | 28.88 | Negative | Negative | Negative | Positive |
| GI_MSA_02_280 | Positive | 35.81 | Negative | Negative | Negative | Positive |
| GI_MSA_02_271 | Positive | 25.47 | Negative | Negative | Positive | N/A |
| GI_MSA_02_304 | Positive | 26.19 | Negative | Negative | Positive | N/A |
| GI_MSA_02_277 | Positive | 26.91 | Negative | Negative | Positive | N/A |
| GI_MSA_02_114 | Positive | 28.54 | Negative | Negative | Positive | N/A |
| GI_MSA_02_451 | Positive | 31.71 | Negative | Negative | Positive | N/A |
| GI_MSA_02_089 | Positive | 25.88 | Negative | Positive | Positive | N/A |
| GI_MSA_02_108 | Positive | 22.58 | Negative | Positive | Positive | N/A |
| GI_MSA_02_248 | Positive | 29.49 | Negative | Positive | Positive | N/A |
| GI_MSA_02_288 | Positive | 19.05 | Negative | Positive | Positive | N/A |
| GI_MSA_02_361 | Positive | 23.11 | Negative | Positive | Positive | N/A |

[a]N/A, testing result not available as testing was not performed.
[b]—, not applicable

thus representing a cost-effective solution for routine diagnostics of the most common bacterial enteric pathogens.

Our data showed comparable clinical performance between the TAQPATH Enteric Bacterial Select Panel and the BD MAX Enteric Bacterial Panel for the detection of *Shigella*, *Salmonella*, and *Campylobacter,* with PPA and NPA of >96% for all targets. The clinical sensitivity and specificity of the TAQPATH Enteric Bacterial Select Panel were determined to be 100% for *Salmonella* and *Shigella/*EIEC, while for *Campylobacter*, the clinical sensitivity was 100% and the specificity was 98.09%. The clinical sensitivity and specificity were determined based on the gold standard composite reference approach, where when a sample is tested with three tests, the result of at least two of the three tests is considered the true result. However, the limitation of this approach is that if one assay shows a significantly higher analytical or clinical sensitivity for a particular target, it is possible that samples will be positive with that test while negative on the two comparators. In this study, the comparative analysis of the six CE-IVD tests for the detection of *S. sonnei*, *S.* Typhimurium, and *C. jejuni* in contrived samples demonstrated higher analytical sensitivity of the TAQPATH Enteric Bacterial Select Panel as compared to the BD MAX Enteric Bacterial Panel, which is consistent with the observed discordant results in clinical samples. Overall, the analysis of contrived samples for the three targets mentioned above showed similar performance between the molecular assays, with some assays included in the comparison demonstrating somewhat lower analytical sensitivity for individual targets (Fig. 3). Previous studies have compared the performance of other currently available multiplexed molecular assays and showed that these assays are reliable methods to detect enteric bacterial pathogens with similar sensitivity and specificity as shown in this study (25–29).

In the prospective study, we demonstrated that the TAQPATH Enteric Bacterial Select Panel enabled the detection of 34.6% more *Campylobacter* infections that could be confirmed by other molecular tests compared to standard routine culture combined with an antigen test. In addition, the data presented show the impact of the use of antigen tests in addition to stool culture, as without the antigen test, routine stool culture would in fact detect <50% of all *Campylobacter* infections. Although bacterial stool culture has historically been considered the gold standard for the diagnosis of enteric bacterial infections, our results showed that multiplex PCR assays constitute a highly accurate alternative for the detection of *Campylobacter*. Due to a low number of positive samples obtained in this part of the study for *Salmonella* spp. (*n* = 2) and no positive specimens for *Shigella* spp./EIEC, we could not provide a meaningful comparison of the two methods to assess the performance for these organisms. Importantly, in this study, the higher observed sensitivity of multiplex real-time PCR tests as compared to routine stool culture provided accurate negative results (representing high specificity) with a fast turnaround time, which could constitute a significant advantage for clinical decision-making. In line with our observations, several other studies reported better or comparable performance of multiplex real-time PCR tests compared to conventional stool culture and microscopy for the detection of enteric bacterial pathogens (6, 8, 9, 12, 30–36). One prospective study that included 1,552 stool specimens showed that culture had a 30% incorrect result rate for the detection of *Campylobacter* spp. as compared to immunoassays and molecular methods (5). Similarly, a retrospective study that included 183 stool samples reported very good diagnostic performance of multiplex PCR testing as compared to conventional culture for *Campylobacter* spp., *Salmonella* spp., *Shigella* spp., *Yersinia enterocolitica*, and *Shiga*-like toxin-producing *E. coli* O157, where PCR was able to detect 34.5% more pathogens than culture (37). May et al. reported that among the specimens tested by both PCR and culture, 12% of *Salmonella* positive, 36% of *Campylobacter* positive, and 74% of *Shigella*/EIEC positive stools were only detected by PCR (38). The limitations of the multiplex PCR assay approach compared to routine culture include that these tests cannot differentiate between living and dead organisms and that they do not offer antibiotic susceptibility results. However, some organisms, such as *Campylobacter,* are difficult to culture as they require specific environmental

conditions and fresh stool samples to recover viable bacteria for culture. Especially, *Campylobacter upsaliensis* also requires specific culturing conditions, which are often not part of routine diagnostic procedures, thus these infections are often underdiagnosed. This could explain why routine stool culture failed to detect *Campylobacter* spp. in nine samples in our study and other studies cited above. In addition, although we could hypothesize that the four samples showing negative results with BD MAX and culture but positive results with the BIOFIRE FILMARRAY GI Panel and the TAQPATH Enteric Bacterial Select Panel are likely to represent *Campylobacter upsaliensis,* we could not confirm this result with the methods used in this study. These data along with what was previously shown by others (38) further highlight PCR-based methods as a valuable tool when compared to conventional culture testing (38).

Rapid testing for enteric bacterial pathogens is crucial for patient management and controlling the spread of enteric disease. In this context, multiplex real-time PCR assays are a useful tool for clinicians who offer a rapid turnaround time. We showed here that results can be obtained from multiplex real-time PCR tests within 3 hours or less when compared with 2–3 days for the conventional culture method. Others have confirmed this difference and showed that prolonged turnaround times associated with stool specimen culture hinder specific etiologic diagnosis information necessary for case management, infection control, and public health interventions (5–14). In terms of environmental impact, the use of molecular PCR testing in a workflow such as the one used for the TAQPATH Enteric Bacterial Select Panel can be highly reduced, as we showed here that this approach generated 7.2-fold less plastic waste in the laboratory compared to routine stool culture. Similarly to other investigations, our cost analysis revealed that the cost of consumables for multiplex PCR for detecting positive samples for the three most common bacterial pathogens is similar and less labor-intensive than conventional culture (11–13). However, labs would need to perform more detailed analyses on the cost-effectiveness of the two methods, which are also dependent on the prevalence of pathogens detected in the patient populations they serve.

In clinical practice, the choice between molecular diagnostic tests and routine stool culture should be guided by the specific clinical scenario, available resources, and the urgency of results. A pragmatic approach might involve initial testing with molecular methods followed by routine stool culture for antibiotic susceptibility and confirmation of *Salmonella* and *Shigella* serotypes testing if this information is required.

In conclusion, our study showed that compared to routine stool culture, the TAQPATH Enteric Bacterial Select Panel is a cost-effective (with higher clinical sensitivity vs similar costs) and highly accurate method for the detection of most common enteric bacterial pathogens, with a faster turnaround time and a lower environmental impact. In this study, the TAQPATH Enteric Bacterial Select Panel also demonstrated higher or similar clinical and analytical performance when compared to other available CE-IVD real-time PCR-based diagnostic tests. Taken together, these results suggest that implementing molecular diagnostic tests to detect enteric pathogens in clinical practice can generate fast and highly reliable results, which have the potential to significantly impact patient outcomes. From a public health perspective, the adoption of molecular diagnostic tests can significantly enhance disease surveillance and outbreak response, as rapid identification of bacterial enteric pathogens can facilitate prompt interventions, thus preventing further transmission and guiding public health strategies for effective control of outbreaks.

## ACKNOWLEDGMENTS

The authors would like to thank the Thermo Fisher Scientific Product Development team for the design and development of the TAQPATH Enteric Bacterial Select Panel and Kelly Li and Michael Tanner for reviewing and editing the final manuscript.

## AUTHOR AFFILIATIONS

[1]Department of Infectious Diseases, MVZ Labor Dr. Limbach & Kollegen GbR, Heidelberg, Germany

[2]Thermo Fisher Scientific, South San Francisco, California, USA

## AUTHOR ORCIDs

Oceane Sorel http://orcid.org/0000-0002-1129-9373

Ulrich Eigner http://orcid.org/0000-0002-0631-5105

## FUNDING

| Funder | Grant(s) | Author(s) |
| --- | --- | --- |
| Thermo Fisher Scientific (TMO) | | Jasmin Koeffer |
| | | Melissa Kolb |
| | | Oceane Sorel |
| | | Camilla Ulekleiv |
| | | Jelena D. M. Feenstra |
| | | Ulrich Eigner |

## AUTHOR CONTRIBUTIONS

Jasmin Koeffer, Conceptualization, Data curation, Formal analysis, Investigation, Methodology, Writing – review and editing | Melissa Kolb, Conceptualization, Formal analysis, Methodology, Writing – review and editing | Oceane Sorel, Data curation, Formal analysis, Writing – original draft, Writing – review and editing | Camilla Ulekleiv, Methodology, Resources, Writing – review and editing | Jelena D. M. Feenstra, Conceptualization, Data curation, Formal analysis, Funding acquisition, Investigation, Methodology, Project administration, Writing – original draft, Writing – review and editing | Ulrich Eigner, Conceptualization, Data curation, Formal analysis, Funding acquisition, Investigation, Methodology, Project administration, Resources, Supervision, Writing – review and editing

## ETHICS APPROVAL

The study was performed according to the Helsinki Declaration and approved by the Research Ethics Committee of the State Medical Association Baden-Wuerttemberg.

## ADDITIONAL FILES

The following material is available online.

### Open Peer Review

**PEER REVIEW HISTORY (review-history.pdf).** An accounting of the reviewer comments and feedback.

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
