## [Reviewer comments · Microbiology Spectrum]

Microbiology Spectrum

Clinical performance evaluation of TaqPath™ Enteric Bacterial Select Panel for detection of common enteric bacterial pathogens in comparison to routine stool culture and other qPCR-based diagnostic tests

Jasmin Koeffler, Melissa Kolb, Oceane Sorel, Camilla Ulekleiv, Jelena Feenstra, and Ulrich Eigner

Corresponding Author(s): Ulrich Eigner, Labor Limbach

Review Timeline:

Submission Date:	August 24, 2023
Editorial Decision:	September 18, 2023
Revision Received:	October 31, 2023
Accepted:	November 1, 2023

Editor: William Lainhart

Reviewer(s): Disclosure of reviewer identity is with reference to reviewer comments included in decision letter(s). The following individuals involved in review of your submission have agreed to reveal their identity: Stacy White (Reviewer #1)

Transaction Report:

DOI: <https://doi.org/10.1128/spectrum.03172-23>

September 18, 2023

Dr. Ulrich Eigner
Labor Limbach
Studies for Infectious Diseases
Im Breitspiel 15
Heidelberg 69126
Germany

Re: Spectrum03172-23 (Clinical performance evaluation of TaqPath{trade mark, serif} Enteric Bacterial Select Panel for detection of common enteric bacterial pathogens in comparison to routine stool culture and other qPCR-based diagnostic tests)

Dear Dr. Ulrich Eigner:

Link Not Available

Sincerely,

William Lainhart

Journals Department
Reviewer comments:

Reviewer #2 (Comments for the Author):

The study by Koeffler et al does a nice job evaluating various GI multiplex panels against themselves as well as the gold standard - stool cultures. The authors also do a nice job discussing the limitations associated with their study. Below are some questions/comments regarding the study:

Introduction (lines 71-73): " It requires an incubation period of up to 4 days to allow the bacteria to grow to a detectable level before being able to identify the resulting colonies (4)." Four days is a bit of an exaggeration. The majority of enteric pathogens can be detected in 24 hours, with *Campylobacter* culture taking up to 72 hours to complete. Please change 4 days to 3 days (or 72 hours).

Clinical samples (lines 98-113). Can you provide any further demographic information for the clinical samples? Age? Gender? Type of patient (inpatient, outpatient, traveler, refugee, etc.)? Immunocompetent or immunocompromised?

RT-PCR testing of clinical stool samples (lines 124-127). Why was the BD MAX system chosen to compare against the TaqPath? And why was the BioFire system chosen to arbitrate discordant results? Do the TaqPath and BD MAX have similar LODs, whereas the BioFire has a lower LOD in comparison to the other two assays. Please provide an explanation, as the authors appear to have access to multiple GI panels when they evaluated their contrived stool samples. This issue is alluded to in lines 239-240.

It would be interesting to know (if the stool samples are still available), whether genetic sequencing of the stool sample would detect *C. upsaliensis* in the samples that were negative by the BD MAX, but positive by the TaqPath and BioFire assays.

Staff Comments:

Preparing Revision Guidelines

Please return the manuscript within 60 days; if you cannot complete the modification within this time period, please contact me. If you do not wish to modify the manuscript and prefer to submit it to another journal, please notify me of your decision immediately so that the manuscript may be formally withdrawn from consideration by Microbiology Spectrum.

Clinical performance evaluation of TaqPath Enteric Bacterial Select Panel for detection of common enteric bacterial pathogens in comparison to routine stool culture and other qPCR-based diagnostic tests

1. Straightforward comparison of this test to others, including conventional.
2. A few things- would be nice to include some details on what Salmonella, Shigella, Ecoli, Campy organisms were used if available (ex. species, serovars and subtype).
3. Not sure if this method would be more cost-effective against routine culture when you include cost of reagents. The author looked at plastic consumption weight differences, but I am not sure what exactly was looked at (pipette tips, transfer pipets, 96 well plates vs petri dish?). This should be clearer, especially if including it as a cost-effective measure. Also, this test, unlike Biofire or BD Max is more complex in performance as it has more manual steps (including extraction) and that does require more skillset to perform. I don't know if this was why the authors decided to focus on comparing implementation to routine stool culture, but I think there is more to consider in the comparison. I am not sure what value this part of the paper really adds.
4. Throughout the document- the term "RT-PCR" is used. This terminology applies strictly to reverse transcriptase (RT) PCR and I don't know if that is being used as "real time", which would be inaccurate. I went to check if this TaqPath assay is an RT-PCR based methodology- and it does not appear to be. Please ensure that if that term is kept that, it is because we are looking at a reverse transcriptase PCR method.

RESPONSE TO REVIEWERS

Response to Reviewer 1:

Comment 1: *Straightforward comparison of this test to others, including conventional.*

Response: We would like to thank the reviewer for taking the time to assess our manuscript.

Comment 2: *A few things- would be nice to include some details on what Salmonella, Shigella, Ecoli, Campy organisms were used if available (ex. species, serovars and subtype).*

Response: Thank you for pointing this out. We agree with this comment. The clinical stool samples we used in this study were leftovers from routine diagnostic testing. These samples were further characterized for serovars following the bacterial stool culture protocol (as described in the material and methods section Lines 149-158). This information is now included in the Methods section (lines 105 to 121). Although the BD MAX™ Enteric and the TaqPath™ Enteric Bacterial Select Panel were designed to detect and differentiate between the most common gastrointestinal bacterial pathogens such as *Salmonella*, *Shigella*/Enteroinvasive *Escherichia coli* (EIEC) and several species of *Campylobacter*, they were not designed to provide differentiated results for each target which is why we decided to simplify the results to the genus of each bacterial target.

For the comparative evaluation of limit of detection we used contrived samples that contained specifically *Campylobacter jejuni*, *Salmonella enterica serovar typhimurium* (*S. typhimurium*) and *Shigella sonnei* (that belongs to serogroup D) (*S. sonnei*) (Lines 160-168).

Comment 3: *Not sure if this method would be more cost-effective against routine culture when you include cost of reagents. The author looked at plastic consumption weight differences, but I am not sure what exactly was looked at (pipette tips, transfer pipets, 96 well plates vs petri dish?). This should be clearer, especially if including it as a cost-effective measure.*

Response: Thank you for pointing this out. We agree with this comment. We have clarified in the methods section that we included all types of laboratory plastic consumables in the calculation. Therefore, we have updated the text to include what types of laboratory plastics were used in the measurement (Lines 186-189). However we did also do the cost analysis based on consumables which is summarized in lines 300-304. While per positive sample the cost amounts to the same considering that the routine culture for positive samples consists of several steps, the cost for negative samples is cheaper for routine culture. For a lab to assess the cost of implementation they would need to look at prevalence of pathogens in their lab, cost of technical time which is much higher for stool culture than for PCR and was not included in the cost analysis and also at the value PCR brings in increasing sensitivity of pathogen detection. This is discussed within the discussion section and we hope is clear.

Comment 4: *Also, this test, unlike Biofire or BD Max is more complex in performance as it has more manual steps (including extraction) and that does require more skillset to perform. I don't know if this was why the authors decided to focus on comparing implementation to routine stool culture, but I think there is more to consider in the comparison. I am not sure what value this part of the paper really adds.*

Response: Thank you for pointing this out. We agree that more parameters should be considered to conduct a proper cost-effectiveness analysis and that it would have been interesting to explore more aspects of this type of analysis. However, given the complexity of such analysis, we decided to focus on three of these parameters which included the cost of consumables per test, plastic consumption and time to result as they can be critical parameters for environmental impact and patient management, respectively. Also, we decided to compare the TaqPath test against routine stool culture because culture remains the gold standard for diagnostics of gastrointestinal infections. We thank the reviewer for this suggestion, and we have updated the text of several sections in the manuscript to reflect the analysis of these two parameters and the potential associated limitations (Lines 184-189 and 389-392).

Comment 5: *Throughout the document- the term "RT-PCR" is used. This terminology applies strictly to reverse transcriptase (RT) PCR and I don't know if that is being used as "real time", which would be inaccurate. I went to check if this TaqPath assay is an RT-PCR based methodology- and it does not appear to be. Please ensure that if that term is kept that, it is because we are looking at a reverse transcriptase PCR method.*

Response: We agree with this comment, the TaqPath assay is a real time PCR method that does not use a reverse transcriptase. We apologize for the confusing abbreviation, and we have incorporated the reviewer's suggestion throughout the manuscript to change "RT" into "Real time" for more clarity. Please see the revised manuscript in attachment. All revisions are marked up using the track changes function for an easy review. We would like to thank the referee again for taking the time to review our manuscript.

Response to Reviewer 2:

Comment 1: *The study by Koeffler et al does a nice job evaluating various GI multiplex panels against themselves as well as the gold standard - stool cultures. The authors also do a nice job discussing the limitations associated with their study. Below are some questions/comments regarding the study.*

Response: We would like to thank the reviewer for taking the time to assess our manuscript.

Comment 2: Introduction (lines 71-73): " It requires an incubation period of up to 4 days to allow the bacteria to grow to a detectable level before being able to identify the resulting colonies (4)." Four days is a bit of an exaggeration. The majority of enteric pathogens can be detected in 24 hours, with Campylobacter culture taking up to 72 hours to complete. Please change 4 days to 3 days (or 72 hours).

Response: We agree with this comment, and we have revised the manuscript accordingly (Lines 71 and 380).

Comment 3: Clinical samples (lines 98-113). Can you provide any further demographic information for the clinical samples? Age? Gender? Type of patient (inpatient, outpatient, traveler, refugee, etc.)? Immunocompetent or immunocompromised?

Response: Thank you for pointing this out. We included the available clinical information in the methods section (lines 105-117). Unfortunately the immune status of patients was not available and therefore this information was not included.

Comment 4: RT-PCR testing of clinical stool samples (lines 124-127). Why was the BD MAX system chosen to compare against the TaqPath? And why was the BioFire system chosen to arbitrate discordant results? Do the TaqPath and BD MAX have similar LODs, whereas the BioFire has a lower LOD in comparison to the other two assays. Please provide an explanation, as the authors appear to have access to multiple GI panels when they evaluated their contrived stool samples. This issue is alluded to in lines 239-240.

Response: The BD MAX test was chosen as the comparator as it has a similar application being a multiplex with only 4 targets, while most of the other tests analyzed in this study are larger multiplex panels. Considering that larger panels might have a lower sensitivity at the expense of higher multiplexing rate, the comparator with similar multiplexing level was chosen for this particular study. The other potential comparator, the RIDA Gene test has previously been used in the clinical study when this product was developed and therefore was not used in this study. Regarding LODs reported within instructions for use of the 2 tests, the LODs for the TaqPath kit are 66CFU/ml for *C. jejuni*, 150CFU/ml for *C. coli*, 25CFU/ml for *C. upsaliensis*, 500CFU/ml for *S. typhimurium*, 500CFU/ml for *S. typhi*, 20CFU/ml for EIEC, 60 CFU/ml for *Shigella* spp., while the LODs reported for the BD MAX test are 296CFU/ml for *S. typhimurium*, 620CFU/ml for *S. enteritidis*, 95CFU/ml for *C. coli*, 42 CFU/ml for *C. jejuni*, 374 CFU/ml for *Shigella flexneri*, 84CFU/ml for *Shigella sonnei*. Based on these values reported in the respective IFUs the 2 tests were deemed to have similar expected performance in terms of limit of detection in clinical samples.

Comment 5: It would be interesting to know (if the stool samples are still available), whether genetic sequencing of the stool sample would detect *C. upsaliensis* in the samples that were negative by the BD MAX, but positive by the TaqPath and BioFire assays.

Response: Although we agree that it would be interesting to confirm that the samples which were negative by the BD MAX, but positive by the TaqPath and BioFire assays would be positive for *C. upsaliensis*; unfortunately, limited remaining sample volume and lack of sequencing capabilities are technical issues which prevent us from doing this analysis. We have however clearly stated this to be a limitation of the study in the discussion section (lines 371-374).

Please see the revised manuscript in attachment. All revisions are marked up using the track changes function for an easy review. We would like to thank the re again for taking the time to review our manuscript.

Re: Spectrum03172-23R1 (Clinical performance evaluation of TaqPath{trade mark, serif} Enteric Bacterial Select Panel for detection of common enteric bacterial pathogens in comparison to routine stool culture and other qPCR-based diagnostic tests)

Dear Dr. Ulrich Eigner:

Your manuscript has been accepted, and I am forwarding it to the ASM production staff for publication. Your paper will first be checked to make sure all elements meet the technical requirements. ASM staff will contact you if anything needs to be revised before copyediting and production can begin. Otherwise, you will be notified when your proofs are ready to be viewed.

Sincerely,
William Lainhart
Editor
Microbiology Spectrum